# Texture Feature Extraction from ^1^H NMR Spectra for the Geographical Origin Traceability of Chinese Yam

**DOI:** 10.3390/foods12132476

**Published:** 2023-06-24

**Authors:** Zhongyi Hu, Zhenzhen Luo, Yanli Wang, Qiuju Zhou, Shuangyan Liu, Qiang Wang

**Affiliations:** 1College of Computer Science and Artifical Intelligence, Wenzhou University, Wenzhou 325035, China; 2Intelligent Information Systems Institute, Wenzhou University, Wenzhou 325035, China; 3Zhenhai District Finance Bureau, Ningbo 315202, China; lzzhen1314@163.com; 4National Health Commission Key Laboratory of Birth Defect Prevention, Henan Institute of Reproductive Health Science and Technology, Zhengzhou 450002, China; hblyfe@163.com; 5College of Chemistry and Chemical Engineering, Xinyang Normal University, Xinyang 464000, China; zhouqiuju@xynu.edu.cn; 6High & New Technology Research Center, Henan Academy of Sciences, Zhengzhou 450002, China; liushuangyan@hnas.ac.cn; 7School of Medicine, Huanghe Science and Technology College, Zhengzhou 450063, China

**Keywords:** local binary pattern, support vector machine, intelligent identification, ^1^H NMR, geographical origin traceability, Chinese yam

## Abstract

Adulteration is widespread in the herbal and food industry and seriously restricts traditional Chinese medicine development. Accurate identification of geo-authentic herbs ensures drug safety and effectiveness. In this study, ^1^H NMR combined intelligent “rotation-invariant uniform local binary pattern” identification was implemented for the geographical origin confirmation of geo-authentic Chinese yam (grown in Jiaozuo, Henan province) from Chinese yams grown in other locations. Our results showed that the texture feature of ^1^H NMR image extracted with rotation-invariant uniform local binary pattern for identification is far superior compared to the original NMR data. Furthermore, data preprocessing is necessary. Moreover, the model combining a feature extraction algorithm and support vector machine (SVM) classifier demonstrated good robustness. This approach is advantageous, as it is accurate, rapid, simple, and inexpensive. It is also suitable for the geographical origin traceability of other geographical indication agricultural products.

## 1. Introduction

Yam (*Dioscorea* spp.) is a popular staple food in Africa, Asia, South America, and the Caribbean [1,2,3,4]. Approximately 600 different yam cultivars have been identified so far [5].Chinese yams (*D. polystachya* Turczaninow) are widely cultivated in East Asia such as China, Korea, and Japan, both as a functional food and for use in traditional Chinese medicine [6,7,8,9].

Traditional Chinese medicine considers geo-authentic herbs that are cultivated in particular locations to have better quality and clinical effects than those grown in other regions [10]. It is acknowledged that *D. polystachya* Turczaninow cultivated in Jiaozuo District (called “Huaiqing” in ancient times, includes Wen County, Mengzhou, Qinyang, Wuzhi, etc.) is the geo-authentic Chinese yam [11,12]. It contains a higher concentration of leucine, threonine, arginine, GABA (gamma-aminobutyric acid), glutamate, aspartate, citrate, and choline than Chinese yam cultivated in Shandong and Shanxi provinces [13]. In addition, the land must lie fallow for several years before the next Chinese yam harvest [14].These properties and prerequisites make the price of geo-authentic Chinese yams much higher than others. As a result, dealers often label Chinese yams from other locations as geo-authentic for higher profits. Adulteration is very common in the herbal industry and seriously restricts the development of traditional Chinese medicine (TCM). Thus [15,16,17,18], accurate geographical origin traceability can effectively ensure drug safety and effectiveness [19].

In our previous study, the distinguishing metabolites between geo-authentic Chinese yams and yams from other locations were confirmed via ^1^H-NMR based metabolomics method [13]. The glucose and fructose content could verify whether or not the yams were geo-authentic. This result was also confirmed by Huang’s group [20]. Although the characteristic metabolites could be clearly distinguished with ^1^H NMR, the content of the metabolites may not be suitable for quantification with this method. Unscrupulous merchants will add exogenous substances to meet the acceptable metabolite content, matching the distinguishing metabolites present in higher levels in geo-authentic herbs. Furthermore, the structural identification of unknown distinguishing metabolites is time-consuming and exhausting work. Therefore, a rapid, simple, inexpensive, and objective identification method is more likely to be accepted.

In geographical origin confirmation, common machine learning algorithms mainly include principal component analysis and linear discriminant analysis [21,22,23,24,25,26,27]. However, the former often ignores the characteristic information contained in components with a low cumulative contribution rate [28]. The latter is not suitable for non-Gaussian distributed samples, which might lead to data overfitting [29]. Using a rotation-invariant uniform local binary pattern (LBP) to extract the discriminant features of data achieves dimensionality reduction, compensating for the limitations of the former statistical analysis methods [30]. The rotation-invariant uniform LBP is a variant of the basic LBP algorithm [31,32]. The basic LBP algorithm has no significant feature extraction effect on high-dimensional data. Moreover, the rectangular neighborhood with a fixed scale cannot effectively capture the ^1^H NMR spectrum local texture features. In addition, because the image rotation transformation cannot obtain the minimum and unique LBP values, the classification model’s performance stability is poor. To solve the above problems, a ring neighborhood is used instead of the rectangular neighborhood based on the LBP algorithm, which can obtain eigenvalues with rotation invariance and quickly achieve dimensionality reduction in high-dimensional data. In addition, to achieve the intelligent identification of yam authenticity, another key point is to choose the appropriate classifier. Common classifiers include k-nearest neighbor (KNN), decision tree (DT), and support vector machine (SVM) [33,34].The performance of SVM is dependent on the selected parameter values. While parameter optimization is time-consuming, SVM’s performance is superior to many other classifiers. In contrast to SVM, KNN has only one parameter; thus, it has the advantage of simple principle and easy operation, better performance, is less time consuming, and is favored by researchers.

In this study, we evaluated an analytical and data analysis approach to achieve the intelligent geographical origin traceability of Chinese yam. A total of three authenticity identification experiments of geo-authentic Chinese yam were carried out, and the relevant experimental performances were analyzed. Firstly, the ^1^H NMR spectra of all yam samples were obtained by nuclear magnetic resonance spectrometer. Then, the texture features of ^1^H NMR images were extracted by rotation-invariant uniform LBP and were input into three classifiers, namely KNN, DT, and SVM, respectively, to evaluate their potential for the authenticity identification of yam samples. This approach is also suitable and can be implemented for the geographical origin traceability of other herbs of commercial and medicinal value.

## 2. Materials and Methods

### 2.1. Plant and Chemical Materials

Geo-authentic Chinese yams (*D. polystachya* cv. Tiegun) were collected in Wen County (E: 112.990, N: 35.005, clay soil, and E: 113.135, N: 34.924, sandy soil), Mengzhou County (E: 112.902, N: 34.918, clay soil; and E: 112.884, N: 34.873, sandy soil), Wuzhi County (E: 113.180, N: 34.988, clay soil; and E: 113.309, N: 34.985, sandy soil), and Qinyang County (E: 112.941, N: 35.026, clay soil), Jiaozuo City, Henan Province.

Non-authentic Chinese yam refers to the yam grown in non-specific producing areas, mainly including *D. polystachya* cv. Jipicao and cv. Xishi (Shandong Province), *D. Polystachya* cv. Xiaobaizui and cv. Ziyao (Hebei Province), *D. polystachya* cv. Tiegun (Shanxi and Shandong Province). The details of the samples were listed in Appendix A.

According to the principle of randomness, the samples were randomly collected. All the samples were authenticated by Zhiming Gao (professor of Henan Agricultural University, Zhengzhou, China) and Xiaolong Xie (associated professor of Henan University of Chinese Medicine, Zhengzhou, China), and voucher specimens were conservedin Henan Academy of Sciences.

Deuterium oxide (D_2_O, 99.9% D with 0.05 wt% TSP, Aldrich) and methanol-*d_4_* (CD_3_OD, 99.8%, CIL) were purchased from Qingdao Tenglong Weibo Technology Co., Ltd. (Qingdao, China). Monosodium phosphate (NaH_2_PO_4_) and disodium phosphate (Na_2_HPO_4_) were purchased from Tianjin Kemiou Chemical Reagent Co., Ltd. (Tianjin, China).

### 2.2. Sample Preparation for ^1^H NMR Analysis

Yam tubers were peeled and cut into slices, then dried with hot air (50 °C) in a drying oven (DZF-6090, Marit Technology, Wuxi, China) for 24–48 h until the sample weights stabilized. Dried yam slices were crushed with an electric grinder (A11BS25, IKA, Germany) and sifted through 60 mesh sieves. Each yam powder (50 mg) was extracted with mixed solvent (0.4 mL phosphate buffer (0.05 mol/L NaH_2_PO_4_ and Na_2_HPO_4_ in D_2_O) and 0.4 mL methanol-d_4_), vortexed for 5 min (MI0101001, Four E’s Scientific, Guangzhou, China), and centrifuged at 10,000× *g* for 5 min (D2012 plus, DragonLab, Beijing, China). Then, the supernatant (550 µL) was transferred into a 5 mm NMR tube for ^1^H NMR analysis.

### 2.3. Acquisition and Processing Parameter

Each ^1^HNMR spectrum was performed on an Agilent 400 MR spectrometer (400 MHz for ^1^H, Agilent Technology, Santa Clara, CA, USA). The acquisition parameters are as follows: probe, 5 mm OneNMR probe; temperature, 298 K; pulse program, PRESAT solvent suppression; number of scans, 64; data points, 64 K; spectral width, 4800 Hz; relaxation delay (d1), 2 s; acquisition time, 3.408 s. All NMR spectra were processed with MestReNova software (version 12.0.1, Mestrelabs Research SL, Santiago de Compostella, Spain).The processing parameters are as follows: window function, exponential; line-broadening, 0.3 Hz. The phase and baseline were corrected manually. The chemical shifts were calibrated with solvent(CD_3_OD at δ 3.31).

### 2.4. Data Preprocessing

For experiment 1, the spectrum was saved as an NMR CSV matrix between the ranges of *δ* 0.50–9.50 for MATLAB analysis (data dimension: 65,536). For experiment 2, the regions of *δ* 3.28–3.34 and *δ* 4.6–5.2 were removed to avoid the effects of the residual signals of solvents (CHD_2_OD and HDO), respectively. Then, the spectrum was saved as an NMR CSV matrix between the ranges of *δ* 0.50–9.50 for MATLAB analysis (data dimension: 45,454). For experiment 3, the spectra between the ranges of *δ* 0.50–9.50 (without *δ* 3.28–3.34 and *δ* 4.6–5.2) were saved as PNG files (1156 px × 600 px)for texture feature capture. Finally, the Matlab software (version 2020a, The MathWorks, Inc., Natick, MA, USA) was used to convert the data into mat format for subsequent experimental algorithm research (data dimension: 10).

In this study, we used the five-fold cross-validation method for unbiased estimation. The data samples were divided into five non-overlapping data subsets in the same proportion, among which the training set accounts for 80% (112 samples), and the rest (28 samples) were used as a test set.

### 2.5. Texture Feature Capture Method

Local Binary Pattern (LBP) was originally proposed by Ojala et al. [32] to be used to describe the texture features of grayscale images. It takes the gray value of the central pixel of the local region as the threshold value to form the coding value of the neighboring pixel by binarization quantization. In general, each neighboring point has a quantization state of 0 or 1. Given that the spatial complexity of the descriptor is proportional to the number of pixel points in the neighborhood *p*, the connection is $2^{p}$. For example, it will generate 256 (2^8^) patterns when *p* = 8. Obviously, as the number of neighborhood sampling sites grows, so does the number of local patterns encoding points, resulting in high-dimensional feature vectors that are dense with redundant information and take up a lot of storage space.

Considering that the dimension of feature vector obtained by LBP algorithm is too high, the binary mode of LBP value will explode with the inin sampling points *p* in the neighborhood [35]. In order to reduce the feature dimension, Ojala et al. extended rotation-invariant LBP to rotation-invariant uniform $LBP_{r,p}^{riu2}$ on the basis of rotation-invariant pattern, and its mathematical expression is as follows [30]:(1)$$LBP_{r,p}^{riu2}=\left\{ \begin{array}{l} \sum_{i=0}^{p-1}s(g_{i}-g_{c}),\;U({LBP}_{r,p})\leq 2\\ p+1,\;\;\;\;otherwise \end{array}\right.$$
(2)$$s(x)=\left\{ \begin{array}{l} 1,\;x\geq 0\\ 0,\;x<0 \end{array}\right.$$
(3)$$U(LBP_{r,p})=|s(g_{p-1}-g_{c})-s(g_{0}-g_{c})|+\sum_{i=1}^{p-1}\left|s(g_{i}-g_{c})-s(g_{i-1}-g_{c})\right|$$
where $U(LBP_{r,p})$ represents the number of conversions of two neighboring values 1 to 0 (or 0 to 1) on a circle with radius *r*. There are *p* + 2 kinds of coding values generated by rotation-invariant uniform LBP, among which there is only *p* + 1 kind of uniform LBP, while all non-uniform patterns are divided into one class and their coding values are represented by one value. As a result of the $LBP_{r,p}^{riu2}$ mode, the feature vector dimension of the entire image is *p* + 2.

In this paper, parameters were set to *r* = 1 and *p* = 8, implying that the $LBP_{1,8}^{riu2}$ algorithm was utilized for feature extraction of ^1^H NMR with a dimension of 10.

### 2.6. Classifiers

#### 2.6.1. K-Nearest Neighbor

K-Nearest Neighbor (KNN) is a machine learning algorithm with simple principle and easy implementation [36] that predicts the labels of test set by summarizing the data points of training set. The basic principle of KNN classifier is to predict the category of data points in the test set based on the majority principle, that is, to predict the category of each test sample by using the category of K known samples that are closest to each other in the feature space.

The mathematical principle of KNN is to calculate the distance between the unknown sample and different feature values, and distinguish the category of the unknown sample according to the distance size. If K-nearest neighbors of the test sample in the feature space all belong to a certain class, the test sample also belongs to this class. Euclidean distance calculation method is selected to measure the feature space distance of sample $X_{1}=[x_{1}^{1},x_{2}^{1},\cdots ,x_{n}^{1}]$ and $X_{2}=[x_{1}^{2},x_{2}^{2},\cdots ,x_{n}^{2}]$, and the calculation formula is as follows:(4)$$d(X_{1},X_{2})=\sqrt{\sum_{i=1}^{n}{\left(x_{i}^{1}-x_{i}^{2}\right)}^{2}}$$

#### 2.6.2. Decision Tree

Decision Tree (DT) [37], as a prediction model, mainly reflects a mapping relationship between object feature values and attributes. On the basis of knowing the probability of occurrence of various situations, the basic principle of this algorithm is to establish decision model by tree structure according to the feature value of data.

When DT is used for classification, the feature vector $x=(x_{1},x_{2},\ldots ,x_{n})$ is input. According to the nodes and paths of the decision tree, new data is mapped to the corresponding category. Its core idea is to gradually segment the data set into different subsets according to the characteristic values of the data until the data of each subset belongs to the same category or reaches the stop condition.

#### 2.6.3. Support Vector Machine

Support Vector Machine (SVM) is the most commonly used classifier to solve classification problems [38]. It is suitable for both supervised machine learning and unsupervised model training. In the case of small sample data, the SVM classifier can achieve high accuracy while taking less time than the standard neural network approach. For the dataset $F=(x_{i},y_{i})$, $i=1,\ldots ,N,\;x\in R^{d},\;y\in \{-1,1\}$, it divides the data into two types viaa hyperplane $f(x)=w^{T}+b$. In order to minimize outliers, the distance between the two types of data is increased, and the geometric interval of the hyperplane is maximized by minimizing *w*. The standard SVM model is defined as follows:(5)$$\left\{ \begin{array}{l} \min(w)=\frac{1}{2}w^{2}+C\sum_{i=1}^{N}\xi_{i}\\ s.t.y_{i}(w^{T}x_{i}+b)\geq 1-\xi_{i},\xi_{i}\geq 0 \end{array}\right.$$
where *w* is the *d*-sustaining number vector perpendicular to the hyperplane, and *b* is the offset from the origin. *C* is the penalty parameter of the classifier and $\xi_{i}$ is the positive relaxation variable.

Kernel functions in LIBSVM software package mainly include linear kernel function, polynomial kernel function, and radial basis function, etc. Experimental results of kernel function selection show that linear kernel function is the best, and linear kernel function does not need optimization parameters. Therefore, linear kernel function is used in this paper.

### 2.7. Evaluation Indicators

A qualitative analysis must be conducted to evaluate the performance of the classification model such as accuracy (ACC), sensitivity (SEN), F1-score (F1), precision (PRE), and recall (REC), among others. In this paper, the first three performance indicators were used to evaluate the classification performance of the above experiments, and the calculation formulas are as follows:(6)$$\{\begin{array}{c} ACC=\frac{TP+TN}{TP+TN+FP+FN}\times 100\%\\ SEN=REC=\frac{TP}{TP+FN}\times 100\%\\ F_{1}=2\cdot \frac{PRE\cdot REC}{PRE+REC} \end{array}$$
where *TP* (True Positive) is the total number of positive samples that are correctly predicted to be positive; *FN* (False Negative) is the total number of positive samples that are incorrectly predicted to be negative; *TN* (True Negative) is the total number of negative samples that are correctly predicted to be negative; *FP* (False Positive) is the total number of negative samples that are incorrectly predicted to be positive.

### 2.8. K-Meansalgorithm

K-means algorithm is an unsupervised learning algorithm that keeps iterating and updating [39]. It divides data samples into different K clusters according to their distance. The smaller the distance between the samples, the higher the probability they classify into one class. The core steps of the algorithm include the following: in the first step, k data points from the samples are randomly selected as the class center, and the square of Euclidean distance between the samples and each class center is calculated. The smallest Euclidean distance value is classified as the same to complete the preliminary clustering effect. The second step is to update the feature mean of each class as a new class center. The above steps are repeated, iterating until the class center does not change and the final clustering result is obtained.

## 3. Results and Discussion

### 3.1. Comparative Analysis of ^1^H NMR Spectra

A total of 140 samples were collected, including 70 geo-authentic Chinese yams and 70 non-authentic Chinese yams, and their ^1^H NMR spectra are shown in Figure 1. By observing the ^1^H NMR spectra characteristics shown in Figure 1a,b, it was found that the overall spectrum shape of geo-authentic and non-authentic Chinese yam was similar, and the corresponding chemical shift of spectral peak was the same. However, significant differences can be observed in the spectral peak intensity. Based on the internal standard TSP, the sugar region (*δ* 6.00–3.50)spectral peak intensity of geo-authentic Chinese yam could be seen as significantly lower than that of non-authentic Chinese yam. The reason for the above phenomenon is that geo-authentic and non-authentic Chinese yam metabolites are the same but differ in terms of concentration. For example, the differences in glucose or amino acid contents result in significant differences in the corresponding signal peak intensity. Geo-authentic medicinal materials are the product of the specific genotype and growth environment interaction. As both geo-authentic and non-authentic Chinese yams are dry roots of *Dioscorea polystachya* Turczaninow, the different varieties and planting areas result in differences in the content of active components, so it is feasible to identify geo-authentic and non-authentic Chinese yams based on the peak intensity of ^1^H NMR spectra.

### 3.2. The Best Method for the Geographical Origin Traceability of Geo-Authentic Chinese Yam

Although the various Chinese yam varieties contain different components, it is difficult to recognize the difference by visual observation.^1^H NMR spectra of geo-authentic and non-authentic Chinese Yyam (Figure 1) revealed significant differences in spectral peak intensity between the two groups. Therefore, machine learning algorithms can be implemented to distinguish the geo-authentic Chinese yam from non-authentic adulterants. In this study, experiment 1 (original spectral data) and 2 (preprocessed spectral data) were carried out to confirm whether the data preprocessing could improve geo-authentic Chinese yam discrimination. Experiments 2 and 3 (features extracted from preprocessed spectral data) were conducted to evaluate whether texture feature extraction based on the rotation-invariant uniform LBP algorithm is further improving model performance (details are listed in the Appendix A).

The corresponding experiments were conducted using the above research methodology, and the experimental results are presented in Table 1. The data preprocessing procedure was crucial because all performance indices in experiment 2 were higher compared to experiment 1. Furthermore, the rotation-invariant uniform LBP algorithm is a novel and effective method for extracting the ^1^H NMR spectrum texture features, as both the original spectra data and the “two-dimensional” spectrum of ^1^H NMR are rich in digital characteristics (experiment 2 and experiment 3). However, obtaining spatial feature information based on the original ^1^H NMR one-dimensional data is challenging. Therefore, the texture features extracted from two-dimensional spectral images by the rotation-invariant uniform LBP algorithm are more conducive to the authenticity identification of geo-authentic Chinese yam. The results of experiments 2 and 3 also confirmed this conclusion. Finally, the performance of SVM and KNN classifiers was better than that of DT, and the classification results obtained by SVM were the best.

Except for the classification accuracy, the speed at which the data are obtained is another important criterion for the judgment of the model. Data preprocessing and spectral texture feature extraction can effectively decrease the feature dimension and the time in the identification process. Therefore, time consumption metrics of three classifiers, namely KNN, DT, and SVM, were also investigated in this study(Figure 2). SVM was the most time-consuming of the three methods. The time consumption of all three classifiers in experiment 3 was approximately equal. Regardless of whether the feature parameters were high- or low-dimensional, the KNN classifier exhibited the lowest time consumed and was the fastest and relatively more stable among the three classifiers. To sum up, suitable classifiers can be selected according to the experimental data dimensionality. For the research conducted in this study, the SVM and KNN classifiers were suitable for geo-authentic Chinese yam identification. SVM is the best choice once rotation-invariant uniform LBP is used to extract the 1H NMR spectra features.

### 3.3. Visualization and Cluster Analysis

To evaluate the feature extraction performance of LBP, a k-means algorithm was carried out to perform a visual analysis of data clustering. Considering the difficulty of high-dimensional data visualization, the mean (Mean) and standard deviation (STD) of each data point were calculated to reduce data dimensionality. The 2D data clustering scatter plots before and after preprocessing are shown in Figure 3. The distance between geo-authentic and non-authentic Chinese yam in Figure 3b was longer than that in Figure 3a, indicating the data preprocessing effect. Although the in-class aggregation effect in Figure 3c is slightly more pronounced compared to Figure 3b, data misclassification is observed in both cases. We observed that the mean and standard deviation of the high-dimensional data only demonstrated the overall data features. However, they disguised the local feature information. Therefore, two features (from a total of ten feature parameters) were randomly selected from LBP features for horizontal and vertical coordinates. The visual clustering scatter plot is presented in Figure 3d. It clearly illustrates that the clustering effect in Figure 3d was better compared to other scatter plots, although only two feature parameters were used to reduce sample division errors. All the above results indicated that the rotation-invariant uniform LBP algorithm was more conducive to geo-authentic Chinese yam identification and discrimination.

### 3.4. Comparison of Different Methods for the Geographical Origin Traceability of Geo-Authentic Chinese Yam

In order to verify herb’s authenticity, various biological and chemical methods are continuously being developed. For example, Yang et al. proposed a rapid and specific polymerase chain reaction (PCR) method using highly yam-specific primers [40].Then, PCR amplification was used to confirm yam authenticity, based on primer amplification and amplicon analysis. Compared with other yams, geo-authentic Chinese yams had the lowest concentrations of glucose and fructose [13]. Thus, their content can function as a distinguishing marker for identifying geo-authentic Chinese yams. The above two methods are based on chemical or biological analysis. Despite the results being accurate, they are not suitable for routine detection because of disadvantages such as being cumbersome, time consuming, and lab or intensive.

Furthermore, some researchers used other analytical instruments along with chemometrics, multivariate statistical analysis, or pattern recognition methods to authenticate geo-authentic Chinese yam. The analytical instruments used were broad-spectrum infrared and near-infrared diffuse reflectance spectroscopy. Their results are listed in Table 2. 

Sun et al. extracted the infrared fingerprint characteristics of yam samples to build a discrimination model, but only reached 70% classification accuracy due to the small data amount [41]. Because of this, Du et al. increased the data amount and constructed a qualitative discrimination model using near-infrared reflectance spectroscopy and discriminant analysis [42]. With this approach, prediction results with 100% accuracy were achieved. Despite the high accuracy achieved by Du et al., [41] the experimental results did not account for the data quantity imbalance in the training process. To obtain a better identification model of geo-authentic Chinese yam and establish a simple and rapid identification method, a five-fold cross-validation method was implemented in this study to avoid data imbalance in the training process. We further implemented an innovative use of the rotation-invariant uniform LBP algorithm to extract the ^1^H NMR images texture features, and with the SVM classifier, eventually achieved 100% identification accuracy. As shown in Table 2, compared with other approaches, the identification model proposed and implemented in this study can achieve more satisfactory results in the authenticity identification problem of geo-authentic Chinese yam.

## 4. Conclusions

In this study, we implemented ^1^H NMR combined with a machine learning algorithm to confirm the geographical origin of geo-authentic Chinese yams. Our results showed that data preprocessing was beneficial in improving identification accuracy. Furthermore, the texture features extraction of ^1^H NMR images based on rotation-invariant uniform LBP algorithm improved the model performance for the identification of geo-authentic Chinese yam. In addition, three different classifiers, KNN, DT, and SVM, were also compared. SVM had the best model recognition performance and reduced time consumption. Finally, the proposed method’s performance was compared to existing approaches. The findings revealed that ^1^H NMR spectra, rotation-invariant uniform LBP, and SVM could successfully identify geo-authentic Chinese yam with 100% identification accuracy. Therefore, this approach can instruct the intelligent identification model of geo-authentic Chinese yams, having multiple advantages such as being quick, accurate, simple, and inexpensive. At the same time, it is also suitable for the geographical origin traceability of other herbs and geographical indications agricultural products.

## Figures and Tables

**Figure 1 foods-12-02476-f001:**
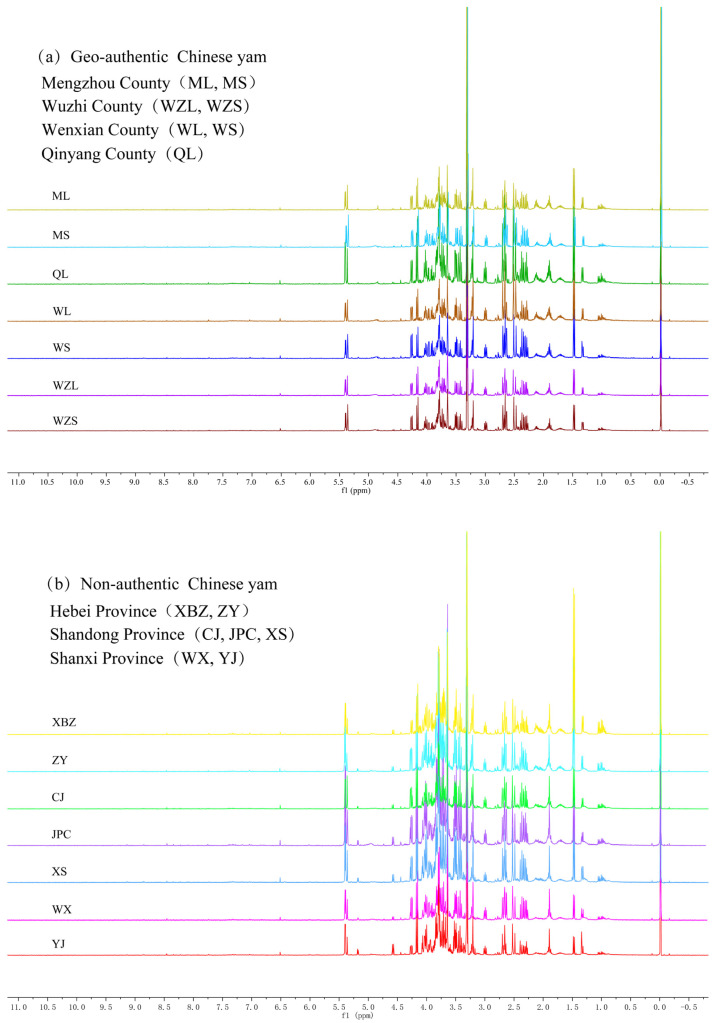
^1^H NMR spectra of geo-authentic Chinese yam (**a**) and non-authentic Chinese yam (**b**).

**Figure 2 foods-12-02476-f002:**
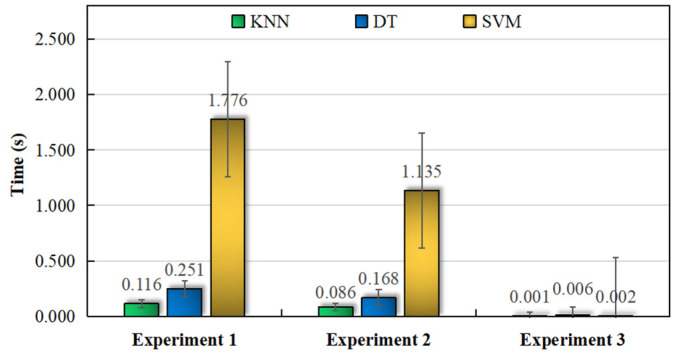
Time consumption among the classifiers for the discrimination of geo-authentic Chinese yam.

**Figure 3 foods-12-02476-f003:**
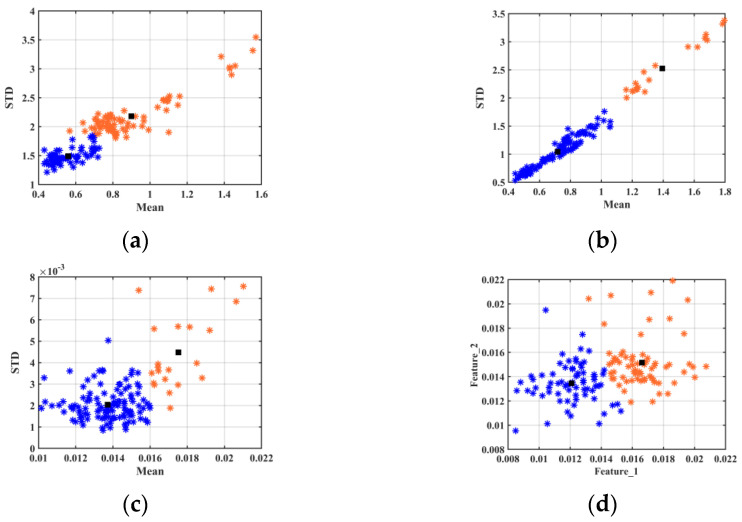
Visual clustering scatter plots of geo-authentic (blue dots) and non-authentic (orange dots) Chinese yam. (Black square: cluster center) (**a**) Clustering effect of the mean and standard deviation on the original data in experiment 1; (**b**) Clustering effect of the mean and standard deviation on the preprocessed data in experiment 2; (**c**) Clustering effect of the mean and standard deviation on the LBP features in experiment 3; (**d**) Clustering effect of the two random parameters on the LBP features in experiment 3.

**Table 1 foods-12-02476-t001:** Identification results of the geo-authentic Chinese yam.

	Classifier	Accuracy (%)	Sensitivity (%)	F1-Score
Experiment 1	KNN	97.14	98.57	0.9719
	DT	97.14	95.71	0.9709
	SVM	97.86	98.57	0.9788
Experiment 2	KNN	99.29	100	0.9931
	DT	97.14	95.71	0.9709
	SVM	99.29	100	0.9931
Experiment 3	KNN	100	100	1
	DT	100	100	1
	SVM	100	100	1

KNN: k-nearest neighbor; DT: decision tree; SVM: support vector machine.

**Table 2 foods-12-02476-t002:** Geo-authentic Chinese yam identification methods and performance comparison.

Authors	Data Size	Methods	Accuracy (%)
Yang et al. [40]	45	Rapid and specific polymerase chain reaction	/
Wang et al. [13]	/	^1^H NMR-based metabolic profiling approach	/
Sun et al. [41]	45	Fourier transform infrared spectroscopy and pattern recognition	70
Du et al. [42]	90	Near-infrared diffuse reflectance spectroscopy and discriminant analysis	100
This work	140	^1^H NMR spectra, rotation-invariant uniform LBP and SVM	100

## Data Availability

The data used to support the findings of this study can be made available by the corresponding author upon request.

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
