# Peer review of "Texture Feature Extraction from 1H NMR Spectra for the Geographical Origin Traceability of Chinese Yam"

_foods, 2023, doi:10.3390/foods12132476_

Round 1
Reviewer 1 Report
Manuscript presents a methodology for the geographical classification of Chinese yams based on H-NMR data by using three different models. Authors applied previously generated LBP algorithm to classify a good amount of dried Chinese yam samples with three different classifiers. There are some suggestions and questions mainly related to some sections in the Materials and Methods section.
Material and Methods:
Line 122. please give the drying time at 50 C, as well.
Line 136: Please state in the text that CHD2OD and HDO are the solvents
Line 144. Please remove the part “...a researcher at university of...” since the reference would be enough for identification.
Lines 215-216: Please explain the meaning of F1-score, PRE and REC terms.
Line 244. Can there be any other fit measurements for the calibration data? What are the R2 values for calibration models?
Results and discussion
Lines 258-261. This paragraph about calibration and validation sets can be moved to “Materials and Methods” section.
Classification of geo-authentic yams is given in Fig.3. What are the number of misclassified samples in calibration and validation sets in a), b), c), and d), or what are the values in Equation 6 used in the calculations of ACC, SEN, F1? They have to be provided.
Author Response
Line 122. please give the drying time at 50 C, as well.
Response: Thank you for your advice, we have added the drying time.
Line 136: Please state in the text that CHD2OD and HDO are the solvents
Response: Thank you for your valuable advice, we have explained them in the text.
Line 144. Please remove the part “...a researcher at university of...” since the reference would be enough for identification.
Response: Thank you for your advice. We have erased it.
Lines 215-216: Please explain the meaning of F1-score, PRE and REC terms.
Response: Thank you for your careful review. F1-score is the harmonic average of precision and recall. Precision (PRE) is the percentage of the predicted positive class was correct. Recall (REC) is the percentage of actual positives our model predicted to be positive.
Line 244. Can there be any other fit measurements for the calibration data? What are the R2 values for calibration models?
Response: Thank you for your careful review. The experiment did not involve regression analysis and calibration (R2) method.
Results and discussion
Lines 258-261. This paragraph about calibration and validation sets can be moved to “Materials and Methods” section.
Response: Thank you for your instructive suggestion. We have adjusted it.
Classification of geo-authentic yams is given in Fig.3. What are the number of misclassified samples in calibration and validation sets in a), b), c), and d), or what are the values in Equation 6 used in the calculations of ACC, SEN, F1? They have to be provided.
Response: Thank you for your careful review. In the manuscript, FIG. 3 is a visual scatter diagram. K-means algorithm is used to prove that extracting texture features is more conducive for yam authenticity identification than classification algorithm. Therefore, ACC, SEN and F1 are not involved in FIG. 3a), b), c) and d).
Reviewer 2 Report
This study investigates of texture feature extraction from 1H NMR spectra for the geographical origin traceability of Chinese yam. The study is well-designed and can address the following major recommendations:
1. Add "Chinese yam" to the keywords.
2. L 30: correct the scientific name of yam: Yams (Dioscorea spp.).
3. L. 94: Add the harvest year as well the study year on these sample.
4. L 114-117: Add the country of the reagents manufacturers.
5. L 119: add the instrument used for drying, its details and model. Also, add the drying time.
6. L 120: add the model of the grinder used. Apply this issue for all instruments used.
7. L 121: add space before mesh.
8. L 123: add models of vortex and centrifuge also edit g to xg.
9. L126: add the country of NMR.
10: L 131: add extra details about data preprocessing.
11. L 257: the figure 1 title is very bad and non informative, so revise it and include part a and part b.
12. the font of Table 1 is non-matched. Also, under the table write the all abbreviated word in complete meaning in the table caption. Apply this issue for all tables and figures.
13. The discussion part with previous studies is very poor (just 3 reports??), add more illustrations based on the related studies.
14. Double check the references style in both text and bibliography according to foods guide.
15. Refer to this study in your study may be in introduction and/or discussion: Metabolome Profiling of Eight Chinese Yam ( Dioscorea polystachya Turcz.) Varieties Reveals Metabolite Diversity and Variety Specific Uses.
https://doi.org/10.3390/life11070687.
Can be improved.
be accepted after
ing
Author Response
- Add "Chinese yam" to the keywords.
Response: Thank you for your valuable advice. We have added it.
- L 30: correct the scientific name of yam: Yams (Dioscoreaspp.).
Response: Thank you for your careful review. We have corrected it.
- L. 94: Add the harvest year as well the study year on these sample.
Response: Thank you for your advice. We have added the experiment date in Table S1 and S2.
- L 114-117: Add the country of the reagents manufacturers.
Response: Thank you for your advice. We have added it.
- L 119: add the instrument used for drying, its details and model. Also, add the drying time.
Response: Thank you for your advice. We have added it.
- L 120: add the model of the grinder used. Apply this issue for all instruments used.
Response: Thank you for your advice. We have added the information.
- L 121: add space before mesh.
Response: We have done.
- L 123: add models of vortex and centrifuge also edit g to xg.
Response: Thank you for your advice. We have done.
- L126: add the country of NMR.
Response: We have done.
10: L 131: add extra details about data preprocessing.
Response: Thank you for your advice. But this already contains all details.
- L 257: the figure 1 title is very bad and non informative, so revise it and include part a and part b.
Response: Thank you for your advice. We have revised it.
- the font of Table 1 is non-matched. Also, under the table write the all abbreviated word in complete meaning in the table caption. Apply this issue for all tables and figures.
Response: Thank you so much for pointing out this issue. We have corrected it.
- The discussion part with previous studies is very poor (just 3 reports??), add more illustrations based on the related studies.
Response: Thank you for your advice. After consulting a lot of literature, there are very few studies on the application of machine learning methods to the authenticity detection of yam, and the lack of model performance indicators makes more studies in the discussion part unable to be carried out. The comparison of previous experiments in this paper is the most representative.
- Double check the references style in both text and bibliography according to foods guide.
Response: Thank you for your careful review. We have done.
- Refer to this study in your study may be in introduction and/or discussion: Metabolome Profiling of Eight Chinese Yam ( DioscoreapolystachyaTurcz.) Varieties Reveals Metabolite Diversity and Variety Specific Uses. https://doi.org/10.3390/life11070687.
Response: Thank you for your advice. We have added the reference in our manuscript.
Round 2
Reviewer 1 Report
Corrections are accepted.
Author Response
Thank you for your review.
Reviewer 2 Report
The manuscript has been revised, however, again, add more details for data preprocessing: you only add few words (All NMR spectra were manually phased, baseline-corrected, and calibrated the shifts), also, what about the other advanced preprocessing techniques like EMSC, MSC etc., in other word, why you choose baseline-correction to your data set?
Additionally, from line 133-155 need a revision, since only line 139-140 are related to this section (data preprocessing), but other lines no, you can add a separate subsection regrading NMR analyses.
Author Response
The manuscript has been revised, however, again, add more details for data preprocessing: you only add few words (All NMR spectra were manually phased, baseline-corrected, and calibrated the shifts), also, what about the other advanced preprocessing techniques like EMSC, MSC etc., in other word, why you choose baseline-correction to your data set?
Response: Thank you for your advice. Phase and baseline correction were applied on the 1H NMR spectra but not the data. I think EMSC, MSC maybe the preprocessing techniques of the data but not the spectra.
Baseline correction of the spectrum is necessary for precise signal integration.
“The baseline of a monodimensional spectrum is the theoretical line which connects the spectrum points which are not peaks (nor artefacts). When this baseline is not flat and/or it is offset from zero, many problems arise. Quantitative measuring of high resolution NMR demands a precise signal integration. These integrals are very sensitive to slight deviations of the spectrum baseline.” Select from “Baseline Correction with MestreNova”
Additionally, from line 133-155 need a revision, since only line 139-140 are related to this section (data preprocessing), but other lines no, you can add a separate subsection regrading NMR analyses.
Response: Thank you so much for your valuable suggestion. We have added the separate subsection.